# From qualitative data to correlation using deep generative networks: Demonstrating the relation of nuclear position with the arrangement of actin filaments

Jyothsna Vasudevan[1,2☯], Chuanxia Zheng[3☯], James G. Wan[4], Tat-Jen Cham[3], Lim Chwee Teck[2,5,6], Javier G. Fernandez[1] *

1 Engineering and Product Development, Singapore University of Technology and Design, Singapore, Singapore, 2 Department of Biomedical Engineering, National University of Singapore, Singapore, Singapore, 3 School of Computer Science and Engineering, Nanyang Technological University, Singapore, Singapore, 4 Engineering Systems and Design, Singapore University of Technology and Design, Singapore, Singapore, 5 Mechanobiology Institute, National University of Singapore, Singapore, Singapore, 6 Institute for Health Innovation and Technology, National University of Singapore, Singapore, Singapore

☯ These authors contributed equally to this work.
* javier.fernandez@sutd.edu.sg

**Data Availability Statement:** Relevant data are within the article and its Supporting Information files. Additional data relevant to this study are

## Abstract

The cell nucleus is a dynamic structure that changes locales during cellular processes such as proliferation, differentiation, or migration, and its mispositioning is a hallmark of several disorders. As with most mechanobiological activities of adherent cells, the repositioning and anchoring of the nucleus are presumed to be associated with the organization of the cytoskeleton, the network of protein filaments providing structural integrity to the cells. However, demonstrating this correlation between cytoskeleton organization and nuclear position requires the parameterization of the extraordinarily intricate cytoskeletal fiber arrangements. Here, we show that this parameterization and demonstration can be achieved outside the limits of human conceptualization, using generative network and raw microscope images, relying on machine-driven interpretation and selection of parameterizable features. The developed transformer-based architecture was able to generate high-quality, completed images of more than 8,000 cells, using only information on actin filaments, predicting the presence of a nucleus and its exact localization in more than 70 per cent of instances. Our results demonstrate one of the most basic principles of mechanobiology with a remarkable level of significance. They also highlight the role of deep learning as a powerful tool in biology beyond data augmentation and analysis, capable of interpreting—unconstrained by the principles of human reasoning—complex biological systems from qualitative data.

## Introduction

Quantitative research methods involve measuring some predefined features of a representative population, gathering numerical data relative to such features, and statistically analyzing the

available from Dataverse at doi:10.7910/DVN/T7HT85 (https://doi.org/10.7910/DVN/T7HT85).

**Funding:** The Singaporean Ministry of Education has supported this research through the MOE2018-T2-2-176 grant to Javier G. Fernandez. The funders had no role in study design, data collection and analysis, decision to publish, or preparation of the manuscript.

**Competing interests:** The authors have declared that no competing interests exist.

data so that it may be generalized to a larger population or explain a particular phenomenon. While the aim of statistical analysis is to remove human bias from the scientific method, feature selection is a purely human process. As researchers, we select those features that can be measured, deem important, or believe useful in hypothesizing research outcomes. There is, however, an implicit preselection, because the possible features are in all cases limited to those we can define or at least conceptualize. In other words, we cannot measure what does not exist for us.

As the selection of measurables is a rational process, the scientific method has hitherto been unavoidably constrained by human interpretation and reasoning [1]. However, this has recently begun to change. Some deep neural networks trained on large datasets are known to develop an intrinsic understanding of images in a way that goes well beyond low-level features, capturing aspects that may not be obvious or conceptualizable in our interpretation of such images [2]. Here, we use that understanding developed by generative networks to interpret raw images of the arrangements of actin filaments in mammalian cells and demonstrate their relationship with the position of the nucleus. This correlation is commonly understood—or intuited—to exist since the mechanical interplay of both structures is known to have a major role in cell activities [3–8] and fate [9, 10], and their relative misplacement is a characteristic of cell malfunction and disease [11–14]. However, demonstrating it with statistical significance is limited by the impossibility of parameterizing the spaghetti-like arrangements of cytoskeletal fibers [15–17]. To avoid such parameterization, we shifted the analysis from the traditional measurement of the distinct features of each substructure (i.e., actin filaments and nuclei) to the isolation of all information related to each substructure in disjoint datasets and the use of the deep generative network to find, without manually created labels or human supervision, a deterministic relationship between the different information sets.

## Results and discussion

The overall structure of the experimental design is presented in Fig 1. To build the paired datasets of nuclei and cytoskeletal fibers, the two substructures were fluorescently tagged with SyTOX Deep Red (660/682) and Alexa-Fluor 488 (490/519), respectively. The selection was based on the lack of overlapping absorptions at the primary emitting wavelengths of helium–neon (HeNe; 633 nm) and argon (Ar; 488 nm) lasers, avoiding any possible crosstalk between fluorophores. Altogether, 4,900 sets of paired images at a resolution of 300 pixels per inch were taken, each set containing an average of 20 cells. The paired dataset was randomly divided into training (80%) and test (20%) images. To find the nuclear position for a given cytoskeletal arrangement, we used a transformer-based architecture [18–20] based on the TFill network [21]. The architecture can be logically divided into three parts: (i) an encoder that takes the image of the cytoskeleton and successively embeds the two-dimensional (2D) image into high-dimensional, low-resolution feature representation; (ii) a transformer utilizes those high-dimensional features to model their dependencies with the high-dimensional features of the nuclei; and (iii) a decoder extracts those learned features in the high-dimensional space and transforms them to an image of nuclei (i.e., back to the low-dimensional, high-resolution space of common images). The results obtained were fed into a discriminator, which evaluated the proximity of the generated nuclei image to the real image and sent the results of that evaluation back to the network, training it further to improve the quality of the generated images. The auxiliary discriminator and the main generator stage a two-players-game, where two networks are trained simultaneously to compete against each other, one to generate increasingly realistic data (i.e., nuclei generation) and the other, the discriminator, improving its ability to differentiate real and generated data [22]. The iteration of this process enabled the network to

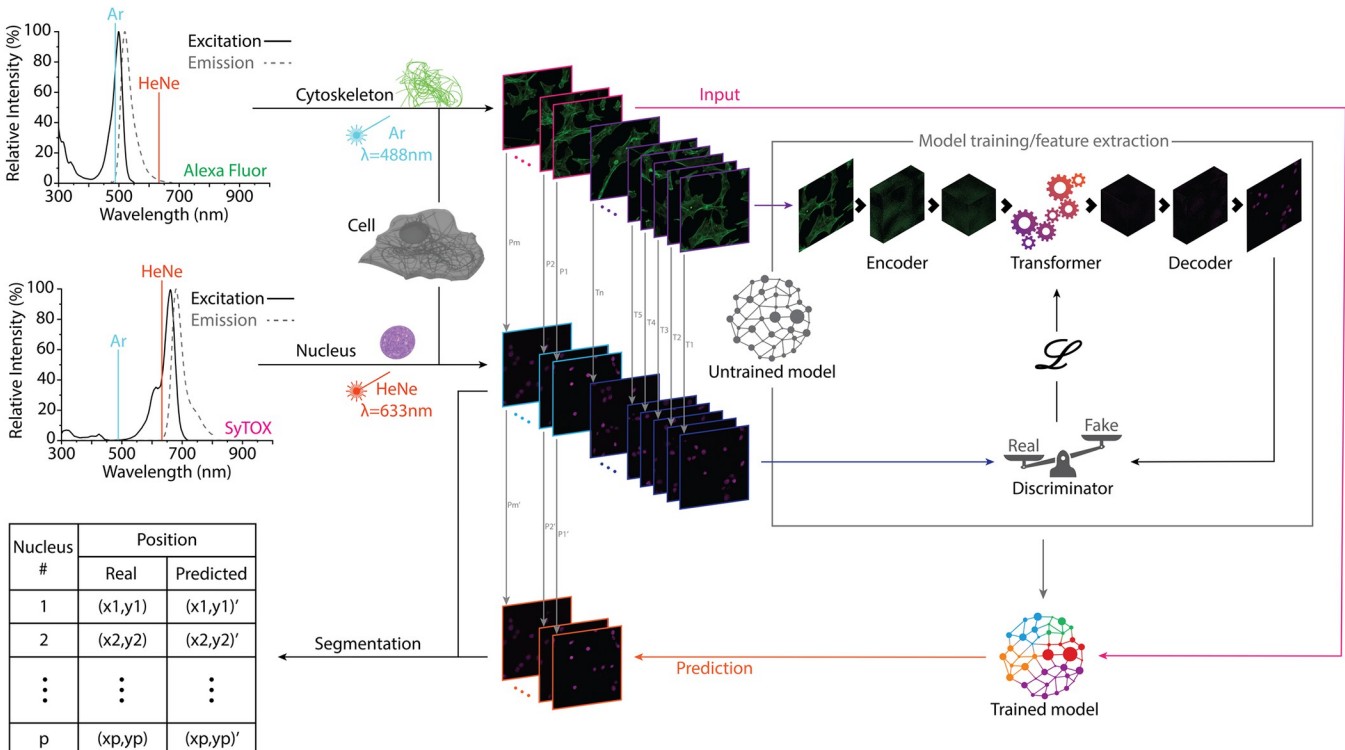

**Fig 1. Demonstration of a correlation between arrangements of actin filaments and nuclear position.** Actin filaments and nuclei information was isolated using non-overlapping fluorophores (Alexa Fluor 488 and SyTOX Deep Red). Then, 80% of the dataset of filaments was used as input to train a transformer-based network using the corresponding paired nuclei to evaluate the proximity to the real solution. The process iteration resulted in a fully trained network that was then used to generate the nuclei of the remaining 20% filament images of the dataset. The generated nuclei and their real counterparts were identified, and the coordinates of their centroids were determined to evaluate the network's ability to predict the nuclear position using only actin filament arrangements.

identify relevant high-dimensional features in the cytoskeleton, enabling successful generation of the associated nucleus. During this process, the network was trained with qualitative data only, in the form of raw microscope images, without interpretation of the images, feature selection, or parameterization. The trained network was then used to generate the nuclei images corresponding to the test images of actin filaments. Thereafter, the images were segmented automatically, extracting the information on the number of nuclei and their diameter and position, and that information was compared with those of the real, or ground truth, nuclei.

The distinctiveness of the network we developed compared to other deep neural networks for image processing is that our objective was not the production of visually convincing images but of images with physiological significance, enabling the demonstration of dependencies between cellular substructures. Therefore, we eliminated image-refining steps aimed at improving appearance, common in other applications, using, for example, only a TFill-Coarse for image-to-image translation during training and testing. The encoder included a block of residual networks (ResNet, Fig 2A), enabling a fast and smooth flow of information across the network by avoiding training for irrelevant layers (i.e., not adding accuracy to the outcome) [23]. The encoded vectors obtained from the input images were then fed into the transformer layer (Fig 2B), whose focus was on accessing the long-range information related to the fibrillar organization in the entire cell. Using a self-attention mechanism, the transformer ensured that all regions of the image, regardless of their location relative to a nucleus, had equal opportunities of flowing through the network's layers [21, 24]. In this way, we prevented the network's bias toward finding an agreement between the predicted information and its immediate

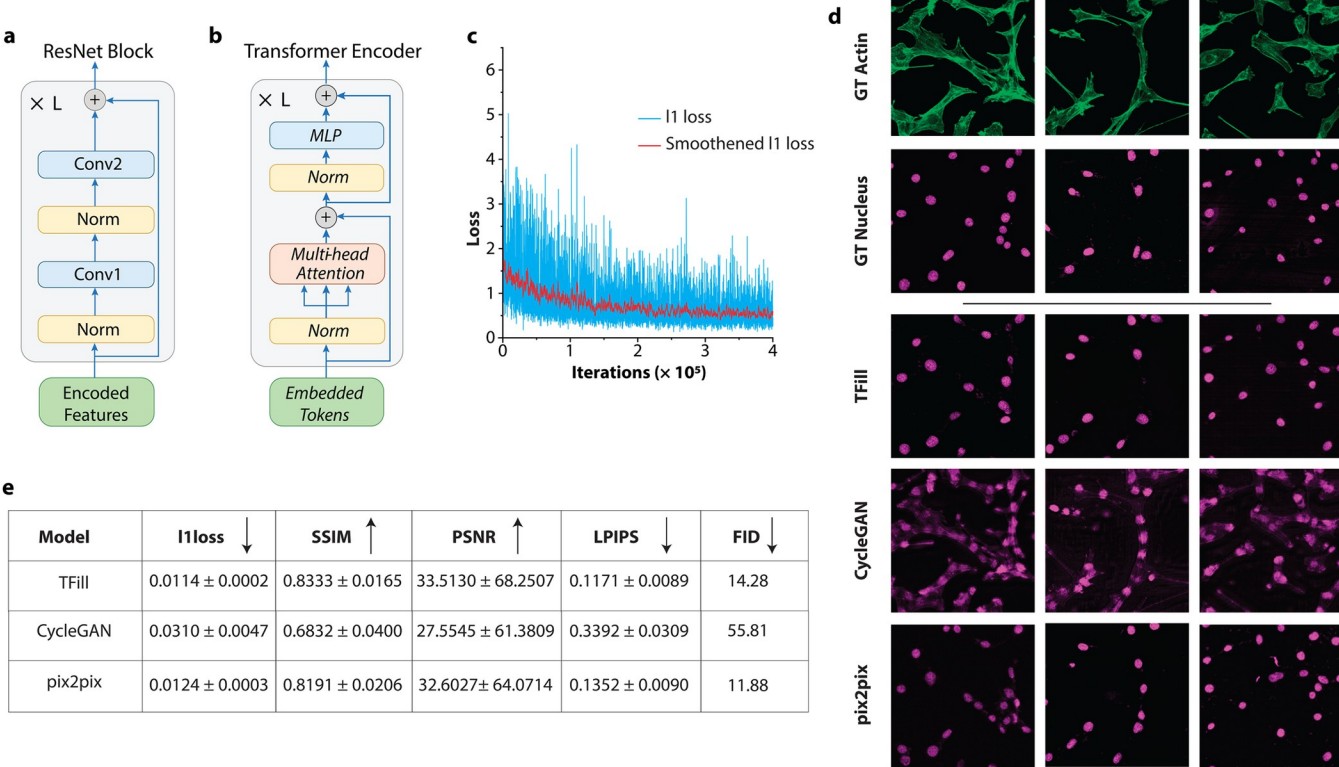

**Fig 2. Assessing the performance of TFill network and comparison with state-of-the-art generative models. a,** Encoder and Decoder layers comprise of a traditional convolutional neural network (CNN)-based ResNet block. **b,** The detailed architecture of the transformer encoder with self-attention mechanism. **c,** Reconstruction loss convergence as a function of iterations. **d,** Visual comparison of TFill generated nuclei images with trending image translation models. **e,** Quantitative comparison of TFill generated images with other image translation models using various metrics from computer vision (↓ Lower is better; ↑ Higher is better).

surrounding (preferred when filling in missing information in photorealistic images) [25] and encouraged agreement of the predicted nuclei with all information related to the fibrillar configuration. Finally, the feature maps were projected back into completed high-resolution images by the decoder and its upsampling layers.

We used two distinctive loss functions during the training process. Since the targeted results, namely the nuclei, were relatively small compared to the background, we used a weighted reconstruction loss to correct the imbalance between the extensive dark background and the small, fluorescent nuclei [26]. This decision was taken after our initial attempts to predict nuclei, which were strongly biased toward the generation of black images, deemed "realistic" by the discriminator because of their proximity to the (mostly black) ground truth. By increasing the weight of the nuclei pixels (by an order of magnitude) in the calculation of the reconstruction loss, the system was encouraged to produce nuclei to achieve acceptable levels of "realism." Then, an adversarial loss function [22] was used to evaluate the proximity of the generated nuclei images to real images of nuclei. This loss function was used and continuously refined by a discriminator trained to spot "fake" (i.e., generated) images of nuclei by comparison with the real images of nuclei. Concurrently, the generator was attempting to fool the discriminator by generating more realistic images while simultaneously minimizing the improving loss function.

The network training converged after $10^5$ iterations (Fig 2C). The trained network was then used to generate the nuclei of the testing dataset. The photographic characteristics of the

results were evaluated on multiple metrics, consisting of pixel-level $\ell_1$ loss curves, region-level Structural Similarity Index (SSIM) [27], and Peak Signal-to-Noise Ratio (PSNR) [28], image-feature-level Learned Perceptual Image Patch Similarity (LPIPS) [29], and dataset-feature-level Fréchet Inception Distance (FID) [30] (Additonal information in the Methods section). Despite the network architecture, which was designed for biological significance to the detriment of photorealism, the generation of images by the developed TFill network was on a par with those generated by state-of-the-art image translation networks geared toward realism, such as CycleGAN and pix2pix (Fig 2D and 2E) [31, 32]. It was not surprising with the structured paired dataset we used that the TFill-based network outperformed CycleGAN, which is conceived for mapping unpaired datasets. More interesting was the comparison with pix2pix, which is also explicitly designed to map paired images but focuses on photorealism [31]. Despite the different aims, both networks produced comparable metrics and realistic images. However, the TFill-based network showed a consistent output irrespective of the number of nuclei in the image, while pix2pix often generated noisy images and a recurrent "mosaic" artifact. Both effects were more frequent in images with many nuclei and were very likely introduced during the upscaling of the images to improve their quality. The performance of TFill (S1 Fig in S1 File) in generating photorealistic images, outdoing networks specifically built for this purpose, highlights the possible use of the presented architecture beyond the aim of this study, namely for predicting fluorescent labels, significantly reducing the processing time of biological samples [33].

In a further step, we evaluated the ability of the developed network to produce scientifically significant data, rather than just realistic images. We extracted the nuclear information using an automated segmentation system to determine and characterize the nuclei of the generated and real nuclei images. We also manually characterized a subset of 120 paired images as quality control for the automated characterization of the complete dataset. Overall, similar results were observed when the counting was done by automatic segmentation and matching or by manual analysis (Fig 3A). In the dataset of real images, 9,659 nuclei were isolated and characterized automatically, while 8,151 (84%) were detected on the synthetic dataset. Similarly, we manually counted 1,359 and 1,069 (i.e., 79%) nuclei in the ground truth and generated datasets, respectively. The 5% difference between automated and manual counting resulted from the different thresholds of complete nucleus used by a human and by the algorithm when counting nuclei at the edges of the images; whereas the human took an educated decision on when to consider a nucleus to fall within an image, the segmentation algorithm tended to detect and count all nuclei partially falling outside the image, resulting in additional counts. On the other hand, the approximately 20% difference between the total real nuclei and those generated by the network resulted from the independent sources of information used to produce the images of ground truth nuclei (i.e., from staining) and generated nuclei (i.e., from actin fibers). Those cases where the information on the actin fiber database was poor or missing resulted in a missing generated nucleus, while the real counterpart was properly stained and identified. This resistance of the network to produce nuclei without enough fiber information is a consequence of the deliberately conservative design of the network, which prioritized quality over quantity, in contrast with the usual "aggressive" approach of image-to-image translation neural networks, where finding a solution (or several) is the priority.

We then demonstrated the deterministic relationship between the nuclear position and the arrangement of actin filaments. We matched each generated nucleus with its real counterpart and calculated the Euclidean distance between their centroids. This was performed automatically by producing a bounding box for each identified nucleus, calculating the overlapping ratio (OR) of each bounding box of a generated nucleus with all those of real nuclei, and pairing them by maximizing the OR (Fig 3B). As before, we kept the manually processed subset as

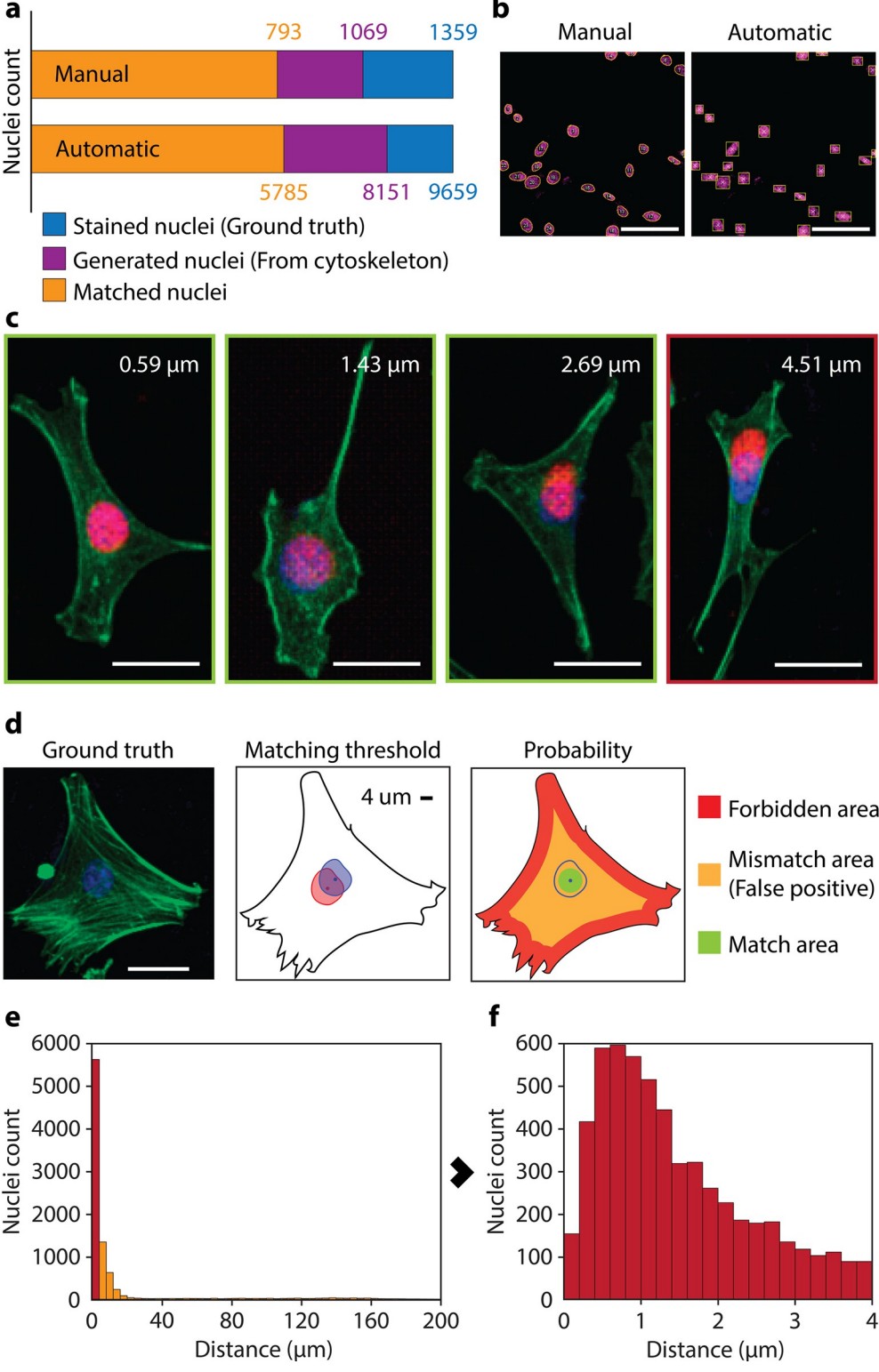

**Fig 3. Positioning of nuclei generated from arrangements of actin filaments. a,** Results of the identification and matching of real and generated nuclei by an automatic counting of the whole generated dataset and by manual counting of a subset of images. Stained nuclei refer to those recorded directly using fluorescent microscopy. Generated nuclei are those produced by the neural network using actin filament arrangements. Matched nuclei are those generated at less than 4 μm of its real counterpart. **b,** Manual (left) and automatic (right) processing of the same image.

In manual processing the profile of the nuclei is drawn to calculate the centroid and the nuclei matched by comparison with the real counterpart. On the other hand, the automatic processing automatically identified the nuclei and generated their bounding boxes, matching generated and real nuclei based on the maximization of the overlapping areas of the bounding boxes. **c,** Several examples of generated nuclei (red) and their corresponding real nuclei (blue). The first three images (green frame) correspond to nuclei generated within the average nuclear radius (4 µm) from their real position. The last image (red frame) corresponds to a mismatch, where the generated nucleus is too far from its real position (See S2 Fig in S1 File for full images; Bars are 20 µm). **d,** Example of a cell and the relative distance of 4 µm within the cytoplasm. The probability of randomly positioning the nucleus within the cytoplasm can be identified as the ratio of possible matched positions for the centroid (green area) with respect to all possible positions (orange area). Those possible positions of the centroid located at less than the nuclear radius from the edges of the cell (red area) are discarded under the premise that the nucleus cannot be positioned partially outside the cell (See S3 Fig in S1 File for further analysis of probabilities; Bar is 20 µm). **e,** Distribution of the distances of the generated nuclei respect their real position. 71% of the nuclei are situated at less than 4 µm of their real position. **f,** Distribution of distances of the generated nuclei considered matched ($<$4 µm). 40% of the matched nuclei are located at less than 1µm from their real position.

control. The bounding boxes were used to also calculate the centroid of each nucleus, which, in the case of generated nuclei, depend on the ability of the neural network to predict the right position and, to a lesser extent, the shape of the nucleus (Fig 3C, S2 Fig in S1 File) [34]. The neural network, using only cytoskeletal information, positioned 71 ± 1% (for a confidence interval (CI) of 95%) of the generated nuclei within the radius of the real nucleus (4 µm), and almost one out of three nuclei (29 ± 1%, for a CI of 95%) were generated at less than 1 µm from the center of the real nucleus (Fig 3D and 3E).

The consensus for biological experiments is to discard the null hypothesis for a p-value of $< 0.05$. In our experiments, the extreme level of confidence makes p values meaningless [35] (Additonal information in the Methods section). For example, given that the system has one out of 500 chances to randomly place the nucleus's centroid at less than 4 µm of the correct position within the image (159.41×159.41 µm), the correct localization of 71% of the nuclei corresponds to a p-value of approximately $10^{-2170}$. Focusing the analysis on the location of the nucleus in the cell, rather than in the image, we would be assuming that the neural network somehow achieved the correct localization of the nuclei by developing: i) an understanding of the low-level characteristics of the filament arrangements (a feature conceptually similar to the limits or shape of the cell); ii) the understanding that the nuclei must be within those limits; and iii) the skills to predict the size and shape of the nuclei (Fig 3D). In such a situation, the problem would reduce in its last instance to the successful localization of the nucleus within the cytoplasm, a task that, in the most optimistic situation, can be randomly achieved in one out of two cases (S3 Fig in S1 File). This restrictive situation results in an equally negligible p = $6.6×10^{-130}$, yielding strong evidence against the null hypothesis (i.e., a random positioning of the nuclei within the cytoskeleton) and demonstrating, with overwhelming significance, one of the most basic principles of cell biology: the correlation between the position of the nucleus and the actin filaments.

## Conclusion

In sum, to demonstrate the correlation between the position of the nucleus and the cytoskeleton in cells, we isolated the information about the substructures using non-overlapping fluorophores and laser lines. We developed an image-to-image translation algorithm based on a TFill network and trained it, using unparameterized images of actin filaments, to extract high-dimensional features relatable to nuclear information. The network's success in accomplishing its task was measured by predicting several thousand nuclei from the arrangements of actin filaments. Seventy-one per cent of the nuclei were generated within the surface of the real nuclei, and almost half of those at less than 1 µm distant from their real position, demonstrating with

astounding significance the hypothesis of a deterministic relation between the arrangements of the actin filaments and the position of the nucleus. This demonstration illustrates the ability to use deep neural networks, outside data analysis or augmentation, as a method to interpret reality beyond the limitations of human conceptualization, and, specifically, to extract features of systems with a complexity unsuitable for quantitative parameterization. Our results also evidence the conditions enabling a transition from a methodology in biology based on human analysis to methods of data acquisition focused on curating information for non-human interpretation.

## Materials and methods

### Cell culture

Mouse fibroblasts (NIH/3T3) (ATCC) were cultured and maintained in Dulbecco's Modified Eagles Medium (high glucose DMEM, Nacalai Tesque) supplemented with 10% (v/v) Fetal Bovine Serum (FBS, Life Technologies) and 1% (v/v) Penicillin-Streptomycin mixed solution (Nacalai Tesque). Cells were incubated in normal physiological conditions (37˚C, 5% $CO_2$), passaged every three days, and their media was replenished every two days.

### Fluorescence labelling

Upon confluence, the cells were trypsinized, centrifuged, and resuspended in fresh DMEM media at a concentration of $1 \times 10^6$ cells/ml. The cells were seeded on cover glass substrates (22 × 22 mm, thickness = 100 μm) at a seeding density of 20000 cells per substrate to ensure adequate spacing between the cells. All substrates were maintained in standard cell culture conditions (37˚C, 5% $CO_2$) for 24 hrs. The cell-seeded glass substrates were rinsed with Dulbecco's Phosphate Buffered Saline (D-PBS) and fixed with 4% (w/v) Paraformaldehyde (PFA) (Merck Millipore) solution in D-PBS for 15 min. Samples were then washed three times (3X), 5 min each time, with D-PBS. Cells were permeabilized with 0.5% (v/v) Triton X-100 in D-PBS for 10 min and then washed 3X, 5 min each time, using D-PBS. Subsequently, the cells were blocked with 3% (w/v) bovine serum albumin (BSA, Gold Biotechnology) for 1 hr. Samples were stained with a 1:500 (v/v) dilution of Alexa-Fluor® 488 phalloidin (Life Technologies) for 30 min and washed 3X, 5 min each time, with D-PBS. Nuclei were stained with a 0.1 μM SyTOX™ Deep Red Nucleic Acid stain (Life Technologies) for 10 min, following which, the substrates were washed thoroughly with D-PBS to remove unbound stains. The coverslips were mounted (Fluoroshield mounting media, Abcam) and stored in the dark until further use.

### Imaging

The images were acquired using a confocal microscope (Zeiss LSM 710) equipped with a 40× lens with numerical aperture 0.95 connected to Zen Black. Coverslips containing fixed cultured cells are mounted on top of a motorized stage. The microscope was programmed to acquire Z-stack images from a 25 x 25 square tile region (5.250 x 5.250 mm), thus acquiring a total of 625 image fields. The motorized stage can translate across x and y directions. The total thickness of each Z-stack was set to 20 μm, and images were acquired at an interval of 1 μm for each field. Two independent channels were acquired: Alexa Fluor 488 Phalloidin (Excitation (max): 495 nm, Emission (max): 518 nm) and SyTOX Deep Red (Excitation (max): 660 nm, Emission(max): 682 nm). During image acquisition, it was ensured that a stitching algorithm was not employed to facilitate the splitting of the large image area. A total of 4900 image pairs were collected across eight samples.

## Dataset preparation

All acquired images were processed using Zen Blue Lite software. Images were first subjected to maximum intensity projection (MIP) to convert the 3D data into a single 2D image. It was ensured that all the features were visible in the single image plane. The large image area was split into two separate channels (Actin and Nuclei) and small, square image tiles, each of size 210 x 210 μm. The images were arranged into folders, each containing one Actin-Nucleus pair image data, using a custom-written Python script. 4900 image pairs were obtained and randomly divided into training (80% of the total dataset) and testing (20% of the total dataset) images.

## Neural network–architecture

Our network architecture extends the TFill network from Zheng *et al*. We only used TFill-Coarse for image-to-image translation since this study focuses more on nuclei positioning than the realistic appearance generation [21]. Its architecture can be logically grouped into three parts: (i) **an encoder** that takes an image $I \in \mathbb{R}^3$ as input and then successively embeds the 2D image into high-dimensional latent space thus yielding a low-resolution token representation $z_0$; (ii) **a transformer** that captures long-range dependencies between encoded token representation and then outputs the global feature representation; (iii) and **a decoder** that takes the learned feature representation and generates all nuclei based on cell shapes. It gradually upsamples the low-resolution feature maps to high-resolution feature maps to achieve the original resolution images.

The transformer architecture was firstly introduced in Natural Language Processing (NLP) and later widely used in various computer vision (CV) tasks, such as scene classification, object detection, instance segmentation, image generation, and translation. A transformer encoder layer consists of a Multihead Self-Attention (MSA) and a Multilayer Perception (MLP) block. The MSA is applied to capture the long-range relationship between each token, while the MLP is responsible for further transforming the merged features from the MSA layers. Furthermore, to achieve the more complex features, the Layernorm (LN) is used before the MSA and MLP block for none-linear projection. They are expressed by:

$$z_0 = [x^1; x^2; \ldots; x^N] + E_{pos} \tag{1}$$

$$z'_l = \mathrm{MSA}(\mathrm{LN}(z_{l-1})) + z_{l-1} \tag{2}$$

$$z_l = \mathrm{MLP}(LN(z'_l)) + z'_l \tag{3}$$

where $z \in \mathbb{R}^{N \times C}$ is the 1D sequence of N tokens $x$ with $C$ channels, $E_{pos} \in \mathbb{R}^{N \times C}$ is the position embedding. $z_0$ is the input sequence of the transformer while $z_l$ is the sequence output in layer $l$.

The components of the neural network are illustrated in detail in Fig 2. Unlike the NLP that naturally treats each word as token [36] embedding for the 1D sequence, the visual image is in 2D without explicit word representation. Following the existing vision transformer [37–39] arch the CNN-based encoder embeds the 2D images as high-dimensional, low-resolution feature representations. The encoder is a ResNet-style convolutional neural network [40]. It is stacked with four repeated residual blocks. Each block consists of two convolutions, each followed by a leaky rectified linear unit with a leakage factor of 0.2 and a pixel-wise normalization. Besides, a skip connection is used to pass the information through a short path quickly. Each block is followed with a learned pooling operation with stride two to halve the resolution of the resulting feature map.

The transformer-based encoder consists of twelve transformer blocks. Each one can automatically capture the long-range dependencies between all locations in the feature maps. Furthermore, instead of the fixed weights in the CNN-based network, the transformer merges information using the input-dependent adaptive weightings, which are decided by the similarities between features. The decoder is an inverse operation of the encoder. It is applied to unfold the high-dimensional, low-resolution features into low-dimensional, high-resolution images. The decoder is also stacked with four repeated residual blocks as in the encoder. However, each block is followed with an upsampling layer to increase the resolution of feature maps. In parallel, the output layer is added after each block to get multilevel, multi-resolution outputs, where the output structure is inspired by StyleGAN v2 [41]. The multi-resolution outputs allow predication errors to quickly backpropagate to the previous encoders that can stabilize and speed up the training. Finally, an auxiliary discriminator using adversarial learning is applied to improve the generated image quality further [22]. We directly adopt the discriminator architecture from the latest StyleGAN v2 [41], which downsamples input images to 4×4 resolution, and then uses fully connected layers to judge them belong to "real(1)" or "fake(0)". This encourages the generated results to match the distribution in the given data.

## Loss functions

**Weighted reconstruction loss.** We first use pixel-weighted reconstruction loss to enable changing the influence of imbalanced nuclei and background ratios in the target image. The loss is defined as:

$$\mathcal{L}_{rec} = \sum_{x \in \Omega} \left\{ \alpha \cdot \left|\left| M(x) \odot (I_{out}(x) - I_{gt}(x)) \right|\right|_1 + \left|\left| (1 - M(x)) \odot (I_{out}(x) - I_{gt}(x)) \right|\right|_1 \right\}$$

where x is pixels in image domain $\Omega$, $I_{out}$ and $I_{gt}$ are the generated nucleus image and the corresponding ground truth respectively, M is the binary mask map where nucleus regions are labeled as "1," and background pixels are labeled as "0". Thus, the first term of the equation is related to the nuclei regions' reconstruction, while the second term is for the background reconstruction. We use the L1 reconstruction loss for each matched pixel in the generated output and ground truth image.

In our images, most pixels belong to the black background. If we rebuild the original ground truth directly, the output will prefer to generate the black image on average. Therefore, we manually increase the weight of nuclei pixels using a factor $\alpha = 10$, as compared with the black background pixels. To do this, we will enforce the model biases to the nuclei regions, resulting in a balance training.

**Adversarial loss.** We further introduce the adversarial loss to encourage the generated nuclei's distribution to be closed to the nuclei's distribution in the ground truth. Following the previous adversarial learning, we model this *minimax game* using an adversarial loss given by:

$$\mathcal{L}_{gan} = \min_G \max_D E_{I_{gt} \sim P_{data}(I_{gt})} \left[ \log D(I_{gt}) \right] + E_{I_{in} \sim P_{data}(I_{in})} \left[ \log(1 - D(G(I_{in}))) \right]$$

where G is the generator and D is the discriminator, and $\mathbf{I}_{gt}$ and $\mathbf{I}_{in}$ are the data from target ground truth sets and input sets. During the training, generator(G) and discriminator(D) parameters are updated alternately. The D is trained to distinguish between generated and ground truth images by maximizing the loss function above. At the same time, the G tries to fool the discriminator to generate more realistic images by minimizing the above loss function.

### Neural network–evaluation

**Fréchet inception distance (FID) [30].**    This metric calculates the mean and variance distance between the feature vectors for ground truth and generated images. A low FID score indicates better performance.

**Learned perceptual image patch similarity (LPIPS) [29].**    This evaluates the diversity of generated images compared to its ground truth, and it is a state-of-the-art metric that correlates to human perceptual similarity. A lower LPIPS score indicates that the generated image is more realistic and similar to the ground truth.

**Structural similarity index (SSIM) [27].**    This metric computes the perceptual distance between a translated image and its corresponding ground truth based on three indices: luminance, contrast, and structure. The higher the SSIM score, the greater the similarity of the two images.

**Peak signal-to-noise ratio (PSNR) [42].**    PSNR calculates the differences in intensity between the ground truth and the generated image and is defined via the mean square error (MSE). A high PSNR score indicates that the intensity of both images is similar.

### Comparison of nuclei count and positioning in real and generated images

The number of nuclei and their positioning were quantified using built-in functions in MATLAB (MathWorks, Natick, MA) [43]. The images were initially subjected to binarization via HSV thresholding. The properties of the nuclei were recorded from the thresholded images, and the nuclei were counted from the **regionprops** function. The function also provides information such as area, centroid, and bounding boxes around the nuclei region. We then filtered the segmented components with a minimum area of 50 pixels to eliminate noise due to thresholding deficiencies. In addition, poorly thresholded images were manually eliminated from further analysis since the goal is not to test for segmentation accuracy. 729 images out of 980 test images were used to calculate errors in nuclei count and positioning accuracy. The percentage error in nuclei detection was computed as follows:

$$\text{Error (\%)} = \frac{\left|\text{Nuclei count}_{\text{GT}} - \text{Nuclei count}_{\text{Gen}}\right|}{\text{Nuclei count}_{\text{GT}}} \times 100$$

where Nuclei count$_{\text{GT}}$ indicates the number of nuclei in the ground truth and Nuclei count$_{\text{Gen}}$ indicates the number of nuclei in the generated image. Subsequently, for every generated nucleus, the bounding box resulting from **regionprops** function was compared with that of ground truth to compute the overlap ratio using the MATLAB in-built function **bboxOverlapRatio**. The overlap ratio (OR) is an intersection over union metric (IoU), which was computed as follows:

$$\textit{Overlap Ratio} = \frac{\textit{Area}\left(BB_{GT}\right) \cap \textit{Area}\left(BB_{Gen}\right)}{\textit{Area}\left(BB_{GT}\right) \cup \textit{Area}\left(BB_{Gen}\right)}$$

where BB$_{\text{GT}}$ and BB$_{\text{Gen}}$ indicates the bounding boxes of ground truth and generated nuclei respectively. This is used to match generated nuclei centroids to ground truth nuclei centroids. Then, an error metric is computed as a Euclidean Distance (ED) between the two coordinates as follows:

$$\text{ED} = \sqrt{\left(X_{GT} - X_{Gen}\right)^2 + \left(Y_{GT} - Y_{Gen}\right)^2}$$

We also use the Clopper and Pearson approach to establish 95% confidence intervals for all of the performance metrics in this study [44].

## Distribution and probability of the nuclei position within the image and the cell

Statistical analysis is done by modeling the number of correct predictions made by the neural network using a binomial distribution, with a probability of success p for each prediction. Due to the very high number (n = 8151) of trials, a normal distribution is used to closely approximate the binomial. Based on 5785 correct predictions at the 4um distance, the 95% score confidence interval for p is computed to be 71.0+/-1.0%. Similarly, based on 2328 correct predictions at the 1um distance, the 95% score confidence interval for p is computed to be 28.6 +/-1.0%.

To calculate the network's success in predicting the correct position of the nuclei, we first calculated how successful any system would be doing it randomly. To do so, we define the ratio of "right positions" and "possible positions" on the area where the nucleus is generated (S3 Fig in S1 File). The correct positions are those within the nucleus area, taking 4 μm as the average radius for the cells employed in this study. This assumption is made based on the possible divergence of the centroid due to imaging and the different shapes of the real and generated nuclei. We calculated the ratio within the whole image as well as within the cell. To provide a realistic approximation and avoid an overrepresentation of possible positions, we impose the additional restriction that the whole nucleus must be inside the cell for the calculation. Therefore, those possible positions situated at a distance of less than a nuclear radius from the edges of the cell were discarded. To get realistic values, we first modeled cells as a circular entity based on the average radius in a confluent culture. Then, we repeated the same calculation for the images used in this study. In that case, the position ratios are strongly dependent on the cell geometry.

We calculated the ratios and associated p-values for several cells, including the two most extreme examples found. In one end are cells narrowly spread around the nucleus and with the cytoplasm distributed in several protrusions (S3C1 Fig in S1 File), strongly limiting the possible positions of the nucleus to the small central space. On the other extreme are cells uniformly spread and with short or inexistent protrusions (S3E Fig in S1 File), where the possible positions for the nucleus roughly cover the whole cytoplasm. We also repeated the same calculations using a threshold distance of 1 μm between the centroids of the real and predicted nuclei. In that case, the number of matched nuclei drops to almost half; however, the probability of correctly positioning it by chance also decreases dramatically, resulting in similarly low p-values.

All the p values reported in this study are negligible, and it makes no difference to have a p = $10^{-100}$ or $10^{-1000}$, since both are in a range of overwhelming statistical significance. In that situation, we find the confidence interval (reported in the main text) a much more informative way to report an error. Despite the absurdity of these negligible p values, we have included them as a comparison with the current standards in biology to highlight the power of the approach presented here. P values are a standard in biological and biomedical research, fields that struggle to achieve p values < 0.05, a thresholds that hardly ensure reproducibility of the results [45]. Here, however, analyzing a similar system, achieve p values that in the worst scenario are < $10^{-100}$., giving a clear picture of the quality of the demonstration compared to those currently accepted in biomedical studies.

## Supporting information

**S1 File. This contains three additional figures on the performance of the generative network, the generation of nuclei, and the statistical analysis.** The cell image database used in this research is available in the supplementary file "NIH3T3_ImageDataset-

20220519T022044Z-001.zip". The network source code can be found at: https://github.com/JGFermart/NuclearPrediction.
(PDF)

## Author Contributions

**Conceptualization:** Jyothsna Vasudevan, Javier G. Fernandez.

**Data curation:** Jyothsna Vasudevan.

**Formal analysis:** Jyothsna Vasudevan, Chuanxia Zheng, James G. Wan, Javier G. Fernandez.

**Funding acquisition:** Javier G. Fernandez.

**Investigation:** Jyothsna Vasudevan, Chuanxia Zheng, Javier G. Fernandez.

**Methodology:** Jyothsna Vasudevan, Chuanxia Zheng, Javier G. Fernandez.

**Project administration:** Javier G. Fernandez.

**Resources:** Javier G. Fernandez.

**Software:** Jyothsna Vasudevan, Chuanxia Zheng.

**Supervision:** Tat-Jen Cham, Lim Chwee Teck, Javier G. Fernandez.

**Validation:** Chuanxia Zheng, Javier G. Fernandez.

**Visualization:** Javier G. Fernandez.

**Writing – original draft:** Jyothsna Vasudevan, Chuanxia Zheng, Javier G. Fernandez.

**Writing – review & editing:** Jyothsna Vasudevan, Chuanxia Zheng, James G. Wan, Tat-Jen Cham, Lim Chwee Teck, Javier G. Fernandez.

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
