## [Decision Letter · Decision Letter 0]

28 Apr 2022

PONE-D-22-06274Determination of nuclear position by the arrangement of actin filaments using deep generative networksPLOS ONE

Dear Dr. Fernandez,

Thank you for submitting your manuscript to PLOS ONE. After careful consideration, we feel that it has merit but does not fully meet PLOS ONE’s publication criteria as it currently stands. Therefore, we invite you to submit a revised version of the manuscript that addresses the points raised during the review process. Please submit your revised manuscript by Jun 12 2022 11:59PM. If you will need more time than this to complete your revisions, please reply to this message or contact the journal office at plosone@plos.org. Please include the following items when submitting your revised manuscript:A rebuttal letter that responds to each point raised by the academic editor and reviewer(s). You should upload this letter as a separate file labeled 'Response to Reviewers'.A marked-up copy of your manuscript that highlights changes made to the original version. You should upload this as a separate file labeled 'Revised Manuscript with Track Changes'.An unmarked version of your revised paper without tracked changes. You should upload this as a separate file labeled 'Manuscript'.

We look forward to receiving your revised manuscript.

Kind regards,

Florian Rehfeldt

Academic Editor

PLOS ONE

Journal Requirements:

2. Thank you for submitting the above manuscript to PLOS ONE. During our internal evaluation of the manuscript, we found significant text overlap between your submission and the following previously published work: 

- https://www.researchgate.net/publication/335781725_Perspective_Dimensions_of_the_scientific_method

Please revise the manuscript to rephrase the duplicated text, cite your sources, and provide details as to how the current manuscript advances on previous work. Please note that further consideration is dependent on the submission of a manuscript that addresses these concerns about the overlap in text with published work.

"The Singaporean Ministry of Education has supported this research through the MOE2018-T2-2-176 grant

to JGF."

"The Singaporean Ministry of Education has supported this research through the MOE2018-T2-2-176 grant to Javier G. Fernandez.The funders had no role in study design, data collection and analysis, decision to publish, or preparation of the manuscript."

Additional Editor Comments:

Please address thoroughly the concerns raised by the reviewers. In particular the first major point from Reviewer 2 is absolutely crucial.

Reviewers' comments:

Reviewer's Responses to Questions

**Comments to the Author**

1. Is the manuscript technically sound, and do the data support the conclusions?

Reviewer #1: Partly

Reviewer #2: Partly

2. Has the statistical analysis been performed appropriately and rigorously? 

Reviewer #1: Yes

Reviewer #2: Yes

3. Have the authors made all data underlying the findings in their manuscript fully available?

Reviewer #1: Yes

Reviewer #2: No

4. Is the manuscript presented in an intelligible fashion and written in standard English?

Reviewer #1: Yes

Reviewer #2: Yes

5. Review Comments to the Author

Reviewer #1: The paper poses an interesting idea to use a encoder-transformer-decoder network to determine the nuclear positioning from cytoskeletal features. The approach, staining, training, and statistics is sound. Furthermore, raw images and documented code is available for the review.

I want to adress following issues:

Major:

- the autors acknowledge the importance of nuclear position and shape, yet fail to adress the nuclear shape, only using centroid, area and bounding box. The method and paper would greatly benefit from usage of nuclei shape and position in relation to cytoskeleton. Related to this, using the regionprops boundingbox results to calculate overlap of nuclei seems ill advised given nuclear shape differences

- the autors cite references from 1997 till 2012 to justify "mechanical interplay of both structures is known to have a

major role in cell activities[3-6] and fate[7]". There are several papers from 2017-2020 that have investigated this in detail. This papers also refute "correlation between cytoskeleton organization and nuclear position has not, to date, been demonstrated" from the abstract

- P.7 Reference missing. Since this is the reference where "components of the neural network are illustrated in detail", it is a major flaw

- several imaging folders in the ImageDataset are empty, for example Set_1420, Set_1436, Set_2577... The dataset thus contains only 497 sets of actin, nuclei and merged images instead of 611. The autors should either upload this data or remove the folders and correct the numbers given in the paper.

Minor:

- "poorly thresholded images were manually eliminated from further analysis" -> 1/4 of images have been removed, I fail to see how 1/4 could have been "poorly thresholded"?

- P.10 "main text) a much informative way to report an error" -> 'more' missing

- the code is given fully, however I needed to install several packages manually and had to do adjustments. However, I noticed the documentation within the code.

Reviewer #2: The authors of this study trained a neural network to predict location

of the cell nucleus from images of the cell actin cytoskeleton. The trained

system performes remarkably well. I have two major comments and a couple

of minor ones:

Major:

1) From actin images it is clear that the brightest actin arrays simply

alighn with cell edges, which is often the case. So roughly the actin image

outlines the cell perimeter. I am pretty sure the nucleus is located in the

geometric center of the cell, and so its location is only coincidentally

defined by actin; in reality it is defined by the cell shape. The authors

should address this problem somehow; otherwise, this study, though technically

good, is pointless.

2) The NN is not truly tested. Some tests would be very easy: use a few

chemical perturbations, like latrunculin, calyculin etc (there are more than

10 cheaply available). See if nuclear localization will still be predictable

from perturbed actin images without further training.

Minor:

1) there is no cell biological discussion of the vast literature on nuclear

positioning and actin localization; without it, it's hard to judge significance

of the results

2) the paper is written pretty densely; it would be good to see a few main

points described for laymen

6. PLOS authors have the option to publish the peer review history of their article (what does this mean?). If published, this will include your full peer review and any attached files.

Reviewer #1: No

Reviewer #2: No

---

## [Author Response · Author response to Decision Letter 0]

19 May 2022

We want to thank the reviewers for their thoughtful and profound evaluation of our work. We have addressed all the technical comments and suggestions in the new version of the article and provided below a point-by-point response to those comments. However, the reviewers will find that some of the comments related to the results’ significance have not been fully addressed. This is not a neglection of the reviewers’ comments, which are educated and on point, but the result of keeping the article’s original focus —maintaining the scientific aim, despite its abstract nature and complexity, is central for us and one of the main drivers of submitting this work to a journal with a technically focused evaluation process (instead of perceived impact and significance).

As mentioned in our cover letter, our article’s main contribution is not in machine learning or biomechanics. Its main contribution is in epistemology. We believe that with the rise of ML and the development of “intelligent” systems, able to perform an inherently rational task as it is the interpretation of qualitative data, the traditional scientific method focused on human perception and reasoning is, for the first time, at a point requiring its revision to enable novel and less “human-centered” approaches to inquiry complex systems. The demonstration in our article that a correlation can be established without the need for a human interpretation or parametrization of a qualitative representation of the targeted system is an apparent proof of that. It is not arbitrary that our first reference is the “Dimensions of the scientific method” by Eberhard Voit.

We want to remark again that the lack of a proper response to the reviewer’s comments related to the perceived impact in the field of biomechanics must not be understood as contempt for their opinion. In the light of the comments, which centered on the article’s mechanobiological aspects, we realized that we succeeded in focusing the introduction, abstract, and conclusion simultaneously on the experimental design and the results. However, we failed to direct the reader to both aspects in the title. Therefore, in this new version, we have proposed changing the title to “From qualitative data to correlation using deep generative networks: Demonstrating the relation of nuclear position with the arrangement of actin filaments,” which we believe better matches the article’s content.

Review Comments to the Author

Reviewer #1: The paper poses an interesting idea to use a encoder-transformer-decoder network to determine the nuclear positioning from cytoskeletal features. The approach, staining, training, and statistics is sound. Furthermore, raw images and documented code is available for the review.

I want to adress following issues:

Major:

- the authors acknowledge the importance of nuclear position and shape, yet fail to address the nuclear shape, only using centroid, area and bounding box. The method and paper would greatly benefit from usage of nuclei shape and position in relation to cytoskeleton. Related to this, using the regionprops bounding box results to calculate overlap of nuclei seems ill advised given nuclear shape differences

We appreciate the reviewer’s comment. We often discussed the influence of the nuclear shape during the development of this study. We acknowledge the extreme relevance of the nuclear shape in mechanobiology and how it might be even more relevant than nuclear position to understand the triggering of specific cell mechanisms. This, however, was not the objective of this study, and we believe it is outside the scope of this paper. Additionally, our microscopy data, collected specifically for the objective of this study (location), would probably not allow us to perform such analysis due to the poor definition of the nuclear shape. We, however, specifically discuss the influence of the nuclear shape in our data (pages 4 and 5) since the centroid of the nuclei will change depending on the geometry. The consideration of the influence of the nuclear shape is also included in our “sanity check” performed with manually characterized data (Figure 3b, analysis in 3a), where the manually defined profile of the nuclei is compared with the automatically generated bounding boxes. We did not see that influence in the calculation of the centroid. We also highlight in the text the (necessary) prediction of the nuclear shape (not only the position) by the algorithm to achieve such results (page 13).

We appreciate how important it would be for the field to correlate the shape of the nucleus with the cytoskeleton arrangement. However, we believe that analysis is a different study outside the scope of this paper.

- the autors cite references from 1997 till 2012 to justify “mechanical interplay of both structures is known to have a major role in cell activities[3-6] and fate[7]”. There are several papers from 2017-2020 that have investigated this in detail. This papers also refute “correlation between cytoskeleton organization and nuclear position has not, to date, been demonstrated” from the abstract

We acknowledge a bias in our references toward those original studies that set the ground for the cytoskeleton-nucleus interactions. This comment on the lack of recent references agrees with a similar one from Reviewer 2. 

We have revised the citations to include (five) more recent articles and corrected the wording in the abstract to avoid understating the field’s current state. 

- P.7 Reference missing. Since this is the reference where “components of the neural network are illustrated in detail”, it is a major flaw .

We apologize for this error. We revised this article multiple times to avoid typos. However, due to the unusually strict format requirements for the initial submission in PLoS ONE, we had to rearrange several sections at the last minute. Because of that, we missed this broken link to the figure and got the error message instead. We have corrected it, and it now directs the reader to Figure 2. We thank the reviewer for such in detailed evaluation.

- several imaging folders in the ImageDataset are empty, for example Set_1420, Set_1436, Set_2577... The dataset thus contains only 497 sets of actin, nuclei and merged images instead of 611. The autors should either upload this data or remove the folders and correct the numbers given in the paper

We have corrected those. We are not sure why the database was incomplete, but our guess is that we hit some limit in either the file size or the length of the paths during the compression. We have updated the database. Please note that the database file must uncompress into 10Gb of data, while the fully trained model (also provided) is around 27Gb. Thank you for pointing out this major issue.

Dataset: https://www.dropbox.com/s/fqlvjrp5aitd1l6/NIH3T3_ImageDataset-20220519T022044Z-001.zip?dl=0

Fully trained model: https://www.dropbox.com/s/q779s51tqrbtdn4/Jyo%20-%20SUTD_Cell.zip?dl=0

Minor:

- “poorly thresholded images were manually eliminated from further analysis” -> 1/4 of images have been removed, I fail to see how 1/4 could have been “poorly thresholded”?

While the algorithm predicting nuclear position has been developed in-house from scratch, the segmentation system is based on existing tools and algorithms implemented in Matlab. Based on our experience, we are not surprised by a general-purpose thresholding algorithm failing 1 out of 4 times in a real scenario of a heterogeneous collection of microscope images of cells at about 70% convergence. A customized system could improve this performance, but since the current performance of the segmentation is more than enough to support the conclusions, we preferred to use standard tools rather than add a new and unrelated topic to the article, as it is the automatic segmentation of images.

- P.10 “main text) a much informative way to report an error” -> ‘more’ missing

Thanks, we changed it.

- the code is given fully, however I needed to install several packages manually and had to do adjustments. However, I noticed the documentation within the code.

We appreciate the comment. We cleaned, simplified, and documented the code to enable its easy use by others. We appreciate the independent test by the reviewer and the extra mile the reviewer went to test its accessibility to other researchers.

Reviewer #2: The authors of this study trained a neural network to predict location of the cell nucleus from images of the cell actin cytoskeleton. The trained system performes remarkably well. I have two major comments and a couple of minor ones:

Major:

1) From actin images it is clear that the brightest actin arrays simply alighn with cell edges, which is often the case. So roughly the actin image outlines the cell perimeter. I am pretty sure the nucleus is located in the geometric center of the cell, and so its location is only coincidentally defined by actin; in reality it is defined by the cell shape. The authors should address this problem somehow; otherwise, this study, though technically good, is pointless.

We agree on the possible mechanism the AI is using to predict the nucleus. We disagree that the achievement is pointless. As mentioned above, this is unsupervised training. The strategy found by the ML is, therefore, purely based on patterns developed by the ML in isolation from human concepts while working on qualitative data (microscope images). At no point do we provide to the algorithm the concept of cell, shape, filaments, or any other conceptualization of the images. All those concepts, rationalizable by our understanding of those images and others that the ML developed without a counterpart on our understanding of the images, are the key and uniqueness of this study. The ability to perform a non-human parametrization of a complex system and achieve a statistical correlation from qualitative data.

2) The NN is not truly tested. Some tests would be very easy: use a few chemical perturbations, like latrunculin, calyculin etc (there are more than cheaply available). See if nuclear localization will still be predictable from perturbed actin images without further training.

We share the interest in the topic with the reviewer, and we agree on further studies sprouting from this. In particular, the prediction of a disease of a specific cell-state based on nuclear dislocation. However, we believe the evidence strongly supports the paper’s conclusion and that the suggested experiments, while extremely interesting, move in a direction that is not compatible with the focus of this study.

Minor:

1) there is no cell biological discussion of the vast literature on nuclear positioning and actin localization; without it, it’s hard to judge significance of the results

The feedback on the age of the references used is common to both reviewers. We have updated the biomechanical citations to more recent ones, which we believe now give a more comprehensive picture of the field’s state. We appreciate the reviewers’ advice.

2) the paper is written pretty densely; it would be good to see a few main points described for laymen

We acknowledge the density of the article. However, it is worth noting that this article moves from experimental design to mechanobiology to machine learning and statistics. We feel it is impossible to elaborate on each field’s basics for non-experts without transforming the article into a sort of review. For example, for the authors in mechanobiology, the microscopy and biological explanations seem to be very basic, almost trivial when explaining the cytoskeleton. At the same time, the ML learning part seem too steep. We found that those in the field of ML feel the opposite, finding trivial the explanation of the work of an adversarial network and puzzling the explanation of mechanical principles of the cell or the non-overlapping fluorophores.

---

## [Decision Letter · Decision Letter 1]

23 Jun 2022

From qualitative data to correlation using deep generative networks: Demonstrating the relation of nuclear position with the arrangement of actin filaments

PONE-D-22-06274R1

Dear Dr. Fernandez,

We’re pleased to inform you that your manuscript has been judged scientifically suitable for publication and will be formally accepted for publication once it meets all outstanding technical requirements.

Kind regards,

Florian Rehfeldt

Academic Editor

PLOS ONE

Additional Editor Comments (optional):

Reviewers' comments:

Reviewer's Responses to Questions

**Comments to the Author**

1. If the authors have adequately addressed your comments raised in a previous round of review and you feel that this manuscript is now acceptable for publication, you may indicate that here to bypass the “Comments to the Author” section, enter your conflict of interest statement in the “Confidential to Editor” section, and submit your "Accept" recommendation.

Reviewer #1: All comments have been addressed

Reviewer #2: All comments have been addressed

2. Is the manuscript technically sound, and do the data support the conclusions?

Reviewer #1: Yes

Reviewer #2: Yes

3. Has the statistical analysis been performed appropriately and rigorously? 

Reviewer #1: Yes

Reviewer #2: Yes

4. Have the authors made all data underlying the findings in their manuscript fully available?

Reviewer #1: Yes

Reviewer #2: Yes

5. Is the manuscript presented in an intelligible fashion and written in standard English?

Reviewer #1: Yes

Reviewer #2: Yes

6. Review Comments to the Author

Reviewer #1: The autors adressed all comments and made changes to most, and gave reasons to why the other comments were not adressed.

I would have liked to see nuclear positioning inorporated as well as a self-written script used for thresholding instead of a common MatLab skript. However, I can see why the autor deems this is as beyond the scope.

Reviewer #2: appropriate revisions...............................................................................................................................................................

7. PLOS authors have the option to publish the peer review history of their article (what does this mean?). If published, this will include your full peer review and any attached files.

Reviewer #1: No

Reviewer #2: No

---

## [Editor Report · Acceptance letter]

21 Jul 2022

PONE-D-22-06274R1 

From qualitative data to correlation using deep generative networks: Demonstrating the relation of nuclear position with the arrangement of actin filaments 

Dear Dr. Fernandez:

I'm pleased to inform you that your manuscript has been deemed suitable for publication in PLOS ONE. Congratulations! Your manuscript is now with our production department. 

Kind regards, 

on behalf of

Dr. Florian Rehfeldt 

Academic Editor

PLOS ONE